# Ethnic Identity and Well-Being of Andean Indigenous People: The Effect of Individualistic and Collectivist Value Orientations

**DOI:** 10.3390/ijerph18136811

**Published:** 2021-06-25

**Authors:** Andrés Gutiérrez-Carmona, Alfonso Urzúa, Karina Rdz-Navarro

**Affiliations:** 1Department of Nursing, Universidad de Antofagasta, Antofagasta 1240000, Chile; 2School of Psychology, Universidad Católica del Norte, Antofagasta 1240000, Chile; alurzua@ucn.cl; 3Department of Sociology, Universidad de Chile, Santiago 7750000, Chile; rdznavarro@uchile.cl

**Keywords:** well-being, ethnic identity, values, Andean indigenous, individualism, collectivism

## Abstract

The aim of this research was to evaluate the mediating effect of the value orientations of collectivism and individualism on the relationship between ethnic identity and well-being, the latter conceived from the worldview of Andean natives. For this purpose, under an observational and cross-sectional design, 395 Lickan-Antay adults (57% women) living in areas of indigenous development and in two cities in northern Chile were surveyed. We used the Lickan-Antay BLA32 well-being scale, a short version of the Portrait 21 Values Questionnaire to measure individualistic and collectivistic values, and an adapted version of the Ethnic Identity Scale. The results show that ethnic identity had a direct positive effect on all three dimensions of well-being (harmony with community life, ethnic harmony and harmony with nature), and total indirect effects on all five dimensions of well-being, one of them originating mainly from collectivist orientations. Individualistic orientations also showed a positive, though less intense, mediating effect on well-being. We conclude that collectivist and individualistic motivational patterns coexist in Lickan-Antay natives and explain an important part of the relationship between ethnic identity and well-being. Finally, we discuss our results and suggest replication of this study in other ethnic contexts to assess the generalizability of these findings to other native peoples of the Andean region of South America.

## 1. Introduction

In recent years, there has been a growing scientific interest in the study of well-being [1], as hedonic (focused on happiness, pleasure, and pain avoidance) and eudaimonic (with emphasis on personal development, the self-realization and fullness of the human being) approaches that have guided a large part of these studies [2].

Although both approaches recognize that the meaning and content of well-being are influenced by sociocultural, psychological, disciplinary, and/or philosophical factors [3], studies with a cultural emphasis have not been widely developed.

Considering this, and recognizing the particularities of the different worldviews, an incipient line of studies on well-being in indigenous peoples has approached the investigation from ecocentric and holistic approaches, emphasizing community and nature well-being over individual well-being [3,4,5].

For the Andean native peoples of South America, well-being conceives a close relationship and integration between individual, community, and Pachamama (Mother Earth) well-being [6] that is strengthened by cultivating wisdom, perseverance, balance in behavior, and compassion from their worldview [7].

Well-being has been related to material aspects such as having good livestock, money, and the opportunity to provide a good education for your children [8]. Additionally, given the close relationship between indigenous peoples and nature, it is expected that a decrease in the well-being of indigenous people and their communities is related to socio-environmental tensions, deforestation, displacement, and the siege on their ancestral lands [9,10]. The tensions derived from the prevailing colonialist social structure in America also negatively affect the well-being of these peoples [11].

The native Andean people or Lickan-Antay, geographically located in northern Chile, present a good case to study of such well-being. Their definition of well-being integrates the individual, the community, and nature, and considers harmony in society and its ethnic-cultural development [4]. Thus, in the current globalized context, where different cultures and realities coexist and partake in space and time, ethnic identity is positioned as an important variable to consider when studying the well-being of indigenous peoples [12].

Ethnic identity is defined as a part of an individual’s self-concept, and derives from knowledge about belonging to a specific ethnic group and given an evaluative and emotional meaning [13]. Some authors [14,15] maintain that ethnic identity occurs as a conscious and free acceptance of this identity that necessarily involves reflection, the comparison of elements of their identity, and the development of a legitimate sense of belonging. Scientific evidence has shown a positive relationship between ethnic identity and well-being in different ethnic groups [16,17,18].

This relationship could be explained by the positive relationship between group identification and positive affect [19] and/or because a greater sense of identity implies greater integration to the group to which the subject feels they belong that favors the perception of well-being [20]. In turn, the relationship between ethnic identity and well-being could be explained by considering that different ethnic identities can emphasize or restrict certain values and/or value orientations [21] that can undermine or strengthen well-being [22].

Values are a comprehensive system of priorities made up of broad goals of variable importance that underlie and guide the attitudes and behavior of individuals and groups [23]. Schwartz [24] proposes the existence of 10 values (i.e., power, achievement, hedonism, stimulation, self-direction, universalism, benevolence, tradition, conformity, and security), organized in four polarities (i.e., self-transcendence vs. self-promotion; conservation vs. open to change). These values, in turn, can be organized according to the interests of the groups and/or individuals [25], individualistic values (i.e., achievement values, power, self-direction, stimulation, and hedonism), collectivism (i.e., conformity, tradition, and benevolence values), or a mixture of such. A more extensive theoretical development of this theory can be reviewed in Schwartz and Bilsky [26,27].

The relationship between values (organized in polar axes or in relation to interest) and well-being has strong empirical and theoretical support [22,28,29].

The evidence suggests that values emphasizing autonomy, responsibility, and equity tend to be healthier and positively related to higher levels of well-being [4,22]. However, this relationship will depend on the context in which it occurs, as cultural egalitarianism and the human development index some variables that have shown to moderate this relationship [22,30]. For example, in countries with a low human development index (HDI), a negative relationship has been observed between the values of collectivist interests with well-being, and this relationship is positive in countries with a high HDI [22,28]. However, in societies with low cultural egalitarianism, the values of individualistic interests will be more positively related to well-being [22,29]. Hence, it is possible to hypothesize that collectivist and individualistic value orientations could mediate the relationship between well-being and ethnic identity in Andean indigenous peoples characterized by collectivist values.

The Lickan-Antay people, like other Andean native peoples, have been described as a community where collectivist values are emphasized and the community is the basic social unit [31].

However, at present, this community is strongly threatened by an acculturation process intensified by the impact of modernity, the growth of national and international tourist activity, large-scale mining development on their ancestral lands, and the settlement of migrants on their land. These factors have strongly impacted the economy, social relations, traditional activities, and socio-environmental context [32].

With this framework, this study aims to evaluate the mediating effect of collectivist and individualistic value orientations in the relationship between ethnic identity and well-being from the Lickan-Antay worldview of northern Chile. It is expected that ethnic identity will have a positive and significant effect on the different dimensions of Lickan-Antay well-being, mediated by collectivist and individualistic value orientations.

## 2. Materials and Methods

### 2.1. Participants

The study population consisted of adults belonging to the Lickan-Antay (Atacameña) ethnic group in Chile. The Lickan-Antay are an Andean ethnic group who have for over 9000 years inhabited the banks of the Rio Loa and the area surrounding the great Atacama Salt Flat in the North of what is now Chile [32]. Currently, only 12% of them live in their ancestral communities, 88% reside in urban sectors such as the cities of Calama and Antofagasta [33]. The study universe is made up of 30,369 Lickan-Antay indigenous people [33]; understanding by them all inhabitants originating from Lickan-Antay lands, descending from an indigenous parent, and carrying at least one Lickan-Antay last name [34].

To access the participants, exponential non-discriminative snowball sampling was performed. This sampling consisted of choosing initial or seed participants, who contacted us with 3 or 4 people who met the inclusion criteria, and then each of these in turn contacted us with the same number of people, until the required number of participants was reached [35]. The final sample comprised 395 people who completely answered the questionnaires; 42.8% were men and 57.2% women, with an average age of 41.31 years (SD = 17.32). Participants came from the Alto El Loa and Atacama La Grande indigenous development areas, and from the cities of Calama and Antofagasta in Chile.

### 2.2. Instruments

#### 2.2.1. Well-Being Scale on the Lickan-Antay People (BLA32)

The BLA32 scale was developed and psychometrically evaluated by Gutiérrez-Carmona et al. [4] and considers 32 items in a six-point Likert-type response format (1 = strongly disagree; 6 = strongly agree), grouped into five dimensions (i.e., internal harmony, harmony in community life, harmony in society, harmony in ethnic-cultural development, and harmony with nature).

In previous studies [4], the overall reliability of the scale was 0.90, and for the subscales, reliability was reported as approximately 0.91 and 0.97. For our study, the reliability indices were 0.94 for the full scale. In the subdimensions, reliability fluctuated between 0.94 (harmony with ethnic-cultural development) and 0.67 (harmony in society).

#### 2.2.2. Ethnic Identity Scale

Four items extracted and adapted from Smith’s [13] scale of ethnic identity were utilized to measure ethnic identity (i.e., “being Lickan-Antay is very important to you,” “you are aware of your Lickan-Antay roots,” “you feel very united/your ethnicity and culture,” “being Lickan-Antay defines who you are very well”), using a four-level Likert-type response format of agreement (1 = strongly disagree; 4 = strongly agree). High scores reflect a strong and positive orientation toward the reference ethnic group. In our study, the scale scores presented a reliability of 0.83.

#### 2.2.3. Collectivist and Individualistic Value Orientations

Values orientation was measured using a scale subset item of the Shalom H. Schwartz questionnaire (Portrait Values Questionnaire—PVQ21), translated into Spanish by Imhoff and Brussino [36]. The scale was presented in a six-point Likert-type response format (1 = not at all similar to me; 6 = very similar to me).

For this research, only the items that, according to the cultural context of the study population, clearly indicated an orientation towards collectivist or individualistic interests were selected and utilized (e.g., in the dimension of collectivist interests, items such as “I believe that it is important that all individuals in the world are treated equally. I believe that everyone should have the same opportunities in life;” “It is important to me to be loyal to my friends. I fully surrender to those close to me.” Measuring the dimension of individualistic values, items such as the following were utilized: “It is important for me to show my abilities. I want people to admire for me what I do.” “To me, it is important to be a very successful person. I expect that people will recognize my accomplishments.”).

Using the five selected items to measure the dimension of collectivist interests, a reliability of 0.83 was obtained, and with the four items selected to measure the dimension of individualistic interests, a reliability of 0.78 was obtained.

### 2.3. Procedures

The project was reviewed and approved by an ethical committee prior to conducting the study. In turn, the project was presented to Lickan-Antay representatives and leaders and to the board of the Atacameños Peoples Council (Lickan-Antay) to obtain their approval. The instrument was administered individually. Participants received information on study objectives, emphasizing voluntary and anonymous participation. Each participant signed an informed consent form. Throughout the investigative process, it was possible to maintain continuous and reflective communication with different Lickan-Antay people who guided the development of this research, and reviewed and evaluated categories, scales, and procedures utilized.

### 2.4. Statistical Analysis

The reliability analysis for internal consistency (Cronbach’s α) and the descriptive analysis were carried out in the SPSS V.21 (IBM Corp., Armonk, NY, USA) statistical program. The analysis with latent variables was performed with Mplus 7.4 (Muthen & Muthen, Los Angeles, CA, USA) [37].

In the first phase, the measurement models were evaluated by confirmatory factor analysis (CFA) for categorical variables with estimation by weighted least squares adjusted by means and variances (WLSMV) as recommended by the specialized literature [38,39] to guarantee an adequate measurement of the constructs that would guarantee the validity of the subsequent statistical results. To evaluate the fit of the models, the following statistics were utilized: Chi-square (χ^2^), Comparative Fit Index (CFI ≥ 0.95), Tucker–Lewis Index (TLI ≥ 0.90), and the mean quadratic error of approach (RMSEA ≤ 0.06).

From these analyses, the need to make some adjustments to the BLA32 scale was determined. Two items of the harmony dimension in society were eliminated, and cross-factor loadings and correlated errors greater than 0.30 were incorporated [38,39,40], with which an optimal fit of the 5-factor model was achieved (χ^2^ (384) = 756.92, *p* < 0.0001; RMSEA = 0.050, IC 90% RMSEA [0.044; 0.055], *p* (RMSEA ≤ 0.05) = 0.545; CFI = 0.985; TLI = 0.983).

An optimal adjustment was obtained in the two-dimensional value model using the selected items (χ^2^ (23) = 44.837, *p* = 0.0042; RMSEA = 0.049, IC 90% (RMSEA) [0.027; 0.070], *p* (RMSEA ≤ 0.05) = 0.50; CFI = 0.991; TLI = 0.986), with a correlation of 0.535 between the orientation towards collectivist and individualistic interests. Meanwhile, the four-item Ethnic Identity Scale achieved an acceptable adjustment (χ^2^ (2) = 5.873, *p* < 0.0531; RMSEA = 0.070, IC 90% RMSEA [0.000; 0.139], *p* (RMSEA ≤ 0.05) = 0.231; CFI = 0.998; TLI = 0.995).

Subsequently, a structural equation model was estimated that considered direct, indirect, and total effects of ethnic identity on the five dimensions of Lickan-Antay well-being (i.e., internal harmony, harmony in community life, harmony in society, harmony in ethnic development, and harmony with nature) mediated by orientation toward collectivist and individualistic interests.

## 3. Results

### 3.1. Well-Being, Ethnic Identity, and Value Orientations

Responses in all dimensions tended to revolve around values close to the upper limit of the scales utilized. Regarding well-being, all dimensions reached valuations that could be considered high, except for the harmony in society dimension. Conversely, the average score in the orientation toward collectivist interests was higher than that towards individualistic values in the participants. The measures of central tendency and dispersion of the scales utilized in this study are shown in Table 1.

### 3.2. Structural Equation Model

The theoretical model that considered the direct and indirect effects of ethnic identity on well-being and the mediating effect of the two dimensions of the collectivist and individualistic value orientations showed a good fit to the data (χ^2^ (819) = 1824.42, *p* < 0.001; RMSEA = 0.056, IC 90% RMSEA [0.052; 0.059], *p* (RMSEA ≤ 0.05) = 0.003; CFI = 0.963; TLI = 0.959). The results of the model and estimated parameters are shown in Figure 1.

#### 3.2.1. Ethnic Identity and Value Orientations

Ethnic identity showed a positive relationship with the two dimensions of the value orientations; however, its effect on collectivist values was stronger (Std (γ) = 0.372, t = 6.866, *p* = 0.001) than on individualistic values (Std (γ) = 0.184, t = 3.117, *p* = 0.002).

#### 3.2.2. Ethnic Identity and Well-Being Domains

Ethnic identity also showed positive direct effects of high intensities with two of the dimensions of well-being: harmony in ethnic-cultural development (Std (γ) = 0.467, t = 8.512, *p* < 0.001) and harmony with nature (Std (γ) = 0.475, t = 8.590, *p* < 0.001), and positive direct effects of medium-low intensity with the harmony dimension in community life (Std (γ) = 0.217, t = 3.549, *p* < 0.001). No direct effects of ethnic identity were observed on the dimension of internal harmony (Std (γ) = 0.091, t = 1.314, *p* = 0.189) nor in the dimension of social harmony (Std (γ) = 0.026, t = 0.263, *p* = 0.793).

#### 3.2.3. Value Orientations and Well-Being Domains

For their part, individualistic and collectivistic values were directly related to most of the different dimensions of well-being. The orientation toward collectivist interests positively influenced all dimensions of well-being, with weak effects on the dimensions of harmony in community life (Std (β) = 0.127, t = 2.290, *p* = 0.022), harmony in ethnic-cultural development (Std (β) = 0.150, t = 2.874, *p* = 0.004), and harmony in nature (Std (β) = 0.176, t = 3.089, *p* = 0.002), and moderated effects on the dimension of internal harmony (Std (β) = 0.344, t = 5.732, *p* < 0.001) and social harmony (Std (β) = 0.285, t = 3.539, *p* < 0.001). This suggests that as collectivist values increase, they tend to increase well-being. Individualist interests did not influence the dimension of harmony with nature (Std (β) = 0.054, t = 0.998, *p* = 0.318).

However, individualistic values influenced positively and with high intensity in social harmony (Std (β) = 0.500, t = 7.751, *p* < 0.001), internal harmony (Std (β) = 0.268, t = 4.995, *p* < 0.001), community harmony (Std (β) = 0.267, t = 5.433, *p* < 0.001), and harmony in ethnic cultural development (Std (β) = 0.147, t = 3.062, *p* = 0.002), although in these latter cases the effect was less intense.

#### 3.2.4. Value Orientations as Mediators

In the mediation analysis (see Table 2), ethnic identity not only directly influenced the dimensions of harmony in community life, harmony with ethnic and cultural development, and harmony with nature, but also had total indirect effects on the five dimensions of Lickan-Antay well-being through collectivist and individualistic values. This total indirect effect was generally small and positive, indicating that if ethnic identity increased, well-being increased in all five dimensions. When examining the specific effects, we observed the indirect effect of ethnic identity on the dimensions of well-being was mainly from collectivist values, except on social harmony, a situation where collectivist and individualistic values allow an indirect effect of ethnic identity of similar magnitude. Therefore, it can be affirmed that the relationship between ethnic identity and well-being is a direct and indirect positive relationship, mediated mainly by collectivist values.

This indicates that ethnic identity has major effects (around 30% of explained variance) on two dimensions of Lickan-Antay well-being: the dimensions of harmony with cultural development and nature; and moderate effects (between 5% and 10% of explained variance) for the other three dimensions of well-being: internal harmony, community harmony, and social harmony [41].

## 4. Discussion

This study sought to evaluate the mediating effect of value orientations on the relationship between ethnic identity and the different dimensions of Lickan-Antay well-being in northern Chile.

Our results support the central hypothesis of the research (i.e., the collectivist and individualistic value orientations mediate the relationship between ethnic identity and the different dimensions of Lickan-Antay well-being), since total indirect effects of ethnic identity were found on well-being, mainly through collectivist orientations, but also through individualistic ones. As with studies carried out on other ethnic groups [16,17,18], this research confirmed that ethnic identity has a positive effect on well-being. Meanwhile, value orientations exerted a total mediating effect on the relationship between ethnic identity and two dimensions of Lickan-Antay well-being (i.e., internal harmony and social harmony) and a partial mediating effect with the other three dimensions of well-being (i.e., harmony in community life, harmony in ethnic-cultural development, and harmony in nature).This can be explained by considering that the process of development of ethnic identity influences the ascription to certain values, shared by the reference group [13,21] which, in turn, can positively or negatively influence well-being, depending on the context [22].

Similarly, when analyzing the specific indirect effects, the relationship between the variables under study was mainly mediated by collectivist value orientation. A possible explanation is that the Lickan-Antay people, like other Andean people, emphasize community and collectivism [32], which can be related to attitudes (e.g., tolerance) and/or behavior (e.g., help) that strengthen social relationships and positively influence well-being [22]. This has also been reported in studies where collective well-being has been incorporated as one of the variables to be measured, as is the case in this study, where the dimensions associated with well-being have a strong social-community component [42], supporting Schwartz’s hypothesis that values are likely to differ in importance between societies with a more collectivist (communal) social structure and those with a more individualistic (contractual) structure [43].

In this study, individualistic value orientation also had a positive relationship with the Lickan-Antay ethnic identity, which can be explained by considering that individualistic and collectivist motivational patterns can coexist and occur simultaneously in a culture, and not necessarily behave as antagonistic polarities [23,28]. In this case, the relationship between ethnic identity and the orientation towards individualistic interest may be a reflection of the neoliberal and capitalist hegemony that broke into the Lickan-Antay cultural context, influencing its values, and in the strategies of identification and reinterpretation of its ethnic identity as people [4,44].

Due to the total measurement effect of value orientations on the relationship between ethnic identity and two dimensions of Lickan-Antay well-being (i.e., internal harmony and social harmony), only significant direct effects of ethnic identity were observed on the relational dimensions of Lickan-Antay well-being (i.e., harmony in community life, harmony in ethnic-cultural development, and harmony in nature).

This could be based on the theory of self-determination [2], which states that the relational aspect is an innate psychological need of the human being that, when satisfied, positively affects well-being. Thus, a greater ethnic identity produces a greater sense of integration with the reference group [20] and positively impacts the definition and valuation of oneself as a member of a specific ethnic group and as a member of a larger society [13] that can positively influence the relational dimensions of Lickan-Antay well-being.

## 5. Conclusions

This study provides evidence that the collectivist and individualistic value orientations fully and partially explain the relationship between ethnic identity and well-being in Andean people. Although the results of this study are limited to a particular case study, they provide inputs to the theories of well-being, ethnic identity and values—topics that are scarcely addressed as a whole in the context of the south-central Andean region of South America. It is therefore important to evaluate and continue investigating these findings based, for example, on longitudinal studies that provide more theoretical and practical evidence regarding the mediating role of values in the relationship between ethnic identity and well-being in the Lickan-Antay and other native peoples of the South Central Andean zone of South America. One point that we believe can be improved is that we have not incorporated a comparison between Schwartz’s value theory and Andean cultural values, as our measurement was limited only to the values proposed by Schwartz, which have shown evidence of having a relatively universal character; however, it would be interesting to develop a scale that enquires about cultural values in ethnic minorities and how they differ from Western measures for future research.

We want to highlight that this study provides evidence of the relationship between global cultural values and well-being, but in a non-Western culture, and also using a definition of well-being constructed from their own worldview, supporting the idea that regardless of culture, adopting an individualistic or collectivist view of how one relates to the world impacts on personal well-being. We also show evidence that the degree of identification with one’s own ethnicity has a positive impact on this relationship, constituting a protective factor.

## Figures and Tables

**Figure 1 ijerph-18-06811-f001:**
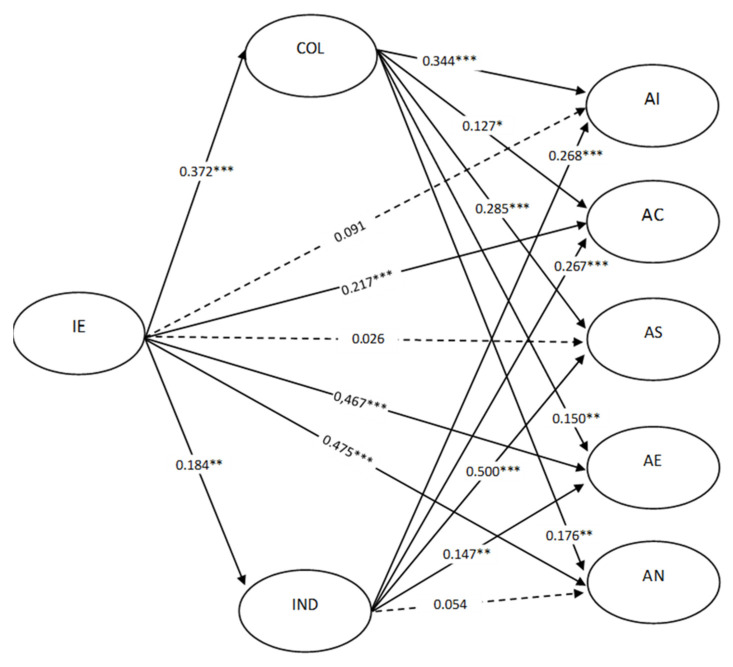
Mediation model using standardized parameters: effect of ethnic identity (IE) in the five dimensions of Lickan-Antay well-being, mediated by the orientation by collectivist (COL) and individualistic (IND) interest. Note: * *p* < 0.05, ** *p* < 0.01, *** *p* < 0.001.

**Table 1 ijerph-18-06811-t001:** Mean and standard deviation for the study scales.

Scale/Subscale	Range	Mean	SD
Lickan-Antay Well-Being			
(a) Internal harmony	1–6	5.12	0.757
(b) Harmony in community life	1–6	4.60	0.920
(c) Harmony in society	1–6	3.15	1.205
(d) Harmony in ethnic-cultural development	1–6	4.91	0.889
(e) Harmony in nature	1–6	5.16	0.850
Ethnic Identity	1–4	3.46	0.592
Orientation towards collectivist interests	1–6	4.91	0.872
Orientation towards individualistic interests	1–6	3.99	1.112

**Table 2 ijerph-18-06811-t002:** Estimates of direct and indirect (total and specific) effects of ethnic identity (IE) on Lickan-Antay well-being.

Effects of the IE	AI	AC	AS	AE	AN
Total (A + B)	0.268 *	0.313 *	0.224 *	0.550 *	0.550 *
(A) Direct total	0.091	0.217 *	0.026	0.467 *	0.475 *
(B) Indirect total (C + D)	0.177 *	0.097 *	0.198 *	0.083 *	0.075 *
(C) Indirect specific collectivism	0.128 *	0.047 *	0.106 *	0.056 *	0.065 *
(D) Indirect specific individualism	0.049 *	0.049 *	0.092 *	0.027 *	0.01

Note: IE = ethnic identity; AI = internal harmony; AC = harmony in community life; AS = social harmony; AE = harmony in ethnic-cultural development; AN = harmony with nature. The estimates presented correspond to standardized parameter values. * indicates a significant relationship at 95% confidence or higher.

## Data Availability

The data presented in this study are available on request from the corresponding author. The data are not publicly available because the project has state funding, and will only be released once the project is finished.

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
