# Peer review of "Ethnic Identity and Well-Being of Andean Indigenous People: The Effect of Individualistic and Collectivist Value Orientations"

_ijerph, 2021, doi:10.3390/ijerph18136811_

Round 1

Reviewer 1 Report

The authors present an interesting study on the relationships between ethnic identity and well-being in the Lickan-Antay ethnic group based on questionnaries with several questions regarding well-being, ethnic identity and collectivist and individualistic value orientations. They conclude that collectivist and individualistic motivational patterns justifies the relationship between ethnic identity and well-being

The research fits with the profile of the journal but in order the manuscript become a paper for IJERPH it needs several changes and corrections.

Abstract: remember that in a abstract there are no references. Take out the three references that appears there. Should then rearrange all the references. At the same time, the abstract is drafted separating objectives, methods, results... for this journal it should be written without this kind of separators. 

Material and methods: I don't understand the block 2.1. I think that is not necessary.  The intentional snowball sampling that at the end yields 395 respondents should be explained in detail. The block instruments should contain subsections with numbers 2.3.1, 2.3.2, 2.3.3,... 

Results: add a frame in fig. 1.  Match paragraph of lines 239-240 with the next (lines 241-245). 

That all, congratulations for the research.

Reviewer 2 Report

  1. Results are not presented clearly in the paper and it is difficult to see how the conclsions are supported by the results.
  2. I would recommend the authors to describe ten basic values more coherently and in conjunction with traditional beliefs and values of Andean native peoples

Reviewer 3 Report

The study is well designed, the methodology is clearly explained and the results are clearly presented. 

Although I find this paper very interesting and well written, I have several suggestions:

  • the literature review should explain intimately the research gap; what is new in this study?
  • the discussion should be extended,
  • authors should more clearly describe their input into science. 

Author Response

Consulte el archivo adjunto.
